


# Icelandic grasslands as long-term C sinks under elevated N inputs

Niki I. W. Leblans[1,2], Bjarni D. Sigurdsson[2], Rien Aerts[3], Sara Vicca[1], Borgthór Magnússon[4] & Ivan A. Janssens[1]

[1] Department of Biology, University of Antwerp, Universiteitsplein 1, 2610 Wilrijk, Belgium
[2] Faculty of Environmental Sciences, Agricultural University of Iceland, Hvanneyri, 311 Borgarnes, Iceland
[3] Department of Systems Ecology , Vrije Universiteit, De Boelelaan 1087, 1081 HV Amsterdam, The Netherlands
[4] Icelandic Institute of Natural History, Urriðaholtsstræti 6–8, P.O. Box 125, 220 Garðabær, Iceland

*Correspondence to*: Niki I. W. Leblans (niki.leblans@uantwerpen.be)

**Abstract.** About 10 % of the anthropogenic CO2 emissions have been absorbed by northern terrestrial ecosystems during the
past decades. It has been hypothesized that part of this increasing carbon (C) sink is caused by the alleviation of nitrogen (N) limitation by increasing anthropogenic N inputs. However, little is known about this N-dependent C sink. Here, we studied the effect of chronic seabird-derived N inputs (47–67 kg N ha$^{-1}$ yr$^{-1}$) on the net soil organic C (SOC) storage rate of unmanaged Icelandic grasslands on the volcanic Vestmannaeyjar archipelago by using a stock change approach in combination with soil dating. We studied both early developmental soils (50 years) and mature soils (1,600 years), and for
the latter we separated between decadal (topsoil) and millennial (total soil profile) responses, where the SOC stocks in the topsoil accorded to 40–50 years of net SOC storage and those in the total soil to 1,600 years of net SOC storage. We found that enhanced N availability - either from accumulation over time, or seabird derived - increased the net SOC storage rate. Under low N inputs, the early developmental soils were weak decadal C sinks (0.018 ton SOC ha$^{-1}$ yr$^{-1}$), but this increased quickly under elevated N inputs to 0.29 ton SOC ha$^{-1}$ yr$^{-1}$, thereby equaling the decadal SOC storage rate of the unfertilized
mature site. Furthermore, at the mature site, chronic N inputs not only stimulated the decadal SOC storage rate, but also the millennial SOC storage was consistently higher at the high N input site. Hence, our study suggests that Icelandic grasslands, if not disturbed, can remain C sinks for many centuries under current climatic conditions and that chronically elevated N inputs can induce a permanent strengthening of this sink.

**Keywords:** Terrestrial C sink, N inputs, Long-term carbon storage, Soil development

# 1 Introduction

The global C cycle plays a prominent role in climate change and is greatly influenced by anthropogenic C emissions (IPCC, 2013). During the past 20 years, terrestrial ecosystems have been absorbing ca. 30 % of the total anthropogenic C emissions; a sink that has been increasing (Le Quere et al., 2009; IPCC, 2013). However, the future evolution of the terrestrial sink-source balance is highly uncertain, and depends on a multitude of factors, such as land use and nutrient dynamics (Poulter et
al., 2011; IPCC, 2013; Fernandez-Martinez et al., 2014; Wieder et al., 2015).





Northern mid- and high latitude terrestrial regions (>50° N) are important C sinks (Ciais et al., 1995; Poulter et al., 2011), accounting for about 30 % of the global net terrestrial C uptake (White et al., 2000). It has been hypothesized that this observed C sink is to a large extent caused by the alleviation of widespread N limitation in these ecosystems (Hudson et al., 1994; Lloyd, 1999; Schlesinger, 2009) due to the three to five fold increase in anthropogenic N deposition during the past

century (Galloway et al., 2008; Gundale et al., 2014). Although a further increase in high-N deposition is projected at northern latitudes (IPCC, 2013), the continuation of the northern C sink is highly debated, with estimates ranging from a current decline to a steady increase until at least the middle of this century (Lloyd, 1999; White et al., 2000; Cramer et al., 2001; Bachelet et al., 2003; Pepper et al., 2005; Friedlingstein et al., 2006; Canadell et al., 2007; Morales et al., 2007; Le Quere et al., 2009; Tao and Zhang, 2010; IPCC, 2013; Todd-Brown et al., 2013; Arora and Boer, 2014). This underlines the

necessity for a better understanding of the N-induced stimulation of long-term C storage in northern ecosystems.

Uncertainties in the further development of this N-dependent northern C sink are related to several aspects. First of all, in spite of the large number of high latitude studies that investigate the short-term effect of N addition on the rates of aboveground C input fluxes (GPP) and ecosystem C output fluxes (litter decomposition and heterotrophic and autotrophic respiration) (Wookey et al., 2009; Bouskill et al., 2014), only a few studies have investigated the effect of long-term N

fertilization on total ecosystem C stocks. Rare studies on this subject report contradicting conclusions, which vary with fertilization rate and litter quality (Hyvonen et al., 2007; Hopkins et al., 2009; Nilsson et al., 2012; Gundersen et al., 2014).

Further, although it is crucial to understand the long-term effects of N on net ecosystem C storage, the few studies have investigated the effect of chronic N inputs on SOC and ecosystem C stocks (>5–8 y) (Hyvonen et al., 2007; Janssens et al., 2010; Leblans et al., 2014) are typically confounded by superficial soil sampling (see e.g. Hobbie et al., 2002; Rumpel and

Kogel-Knabner, 2011; Fornara et al., 2013; Olson and Al-Kaisi, 2015). Soil organic carbon (SOC) dynamics and their responses to N addition are regulated by contrasting mechanisms in the topsoil and the subsoil (Mack et al., 2004; Zehetner, 2010; Appling et al., 2014; Batjes, 2014; Tan et al., 2014). Topsoil decomposition is mainly regulated by nutrient supply, while subsoil decomposition is limited by energy as the proportion of recalcitrant litter increases with depth (Fontaine et al., 2007; Wutzler and Reichstein, 2008). Therefore, focusing only on the upper soil layers can lead to incorrect estimations of

the actual long-term SOC storage.

Finally, while most ecosystems are in an intermediate successional stage as a consequence of former disturbances (Kroël-Dulay et al., 2015), this is often neglected in studies on net ecosystem C storage. Ignoring ecosystem successional and soil developmental stage likely contributes to a substantial part of the uncertainties in global C dynamics (Chapin et al., 2011). For instance, the link between elevated N inputs and net ecosystem C storage might change during the course of soil

maturation (Crocker and Major, 1955; Saynes et al., 2005; Seedre et al., 2011; Appling et al., 2014). This is especially true for N-limited ecosystems (Aerts and Chapin, 2000; Reich and Oleksyn, 2004), where total N stocks and plant available N increase during the soil maturation process (Crocker and Major, 1955; White et al., 2004; Rhoades et al., 2008; Smithwick et al., 2009).



Unmanaged northern grasslands have an extensive coverage (10 % of the global terrestrial surface; Chapin et al., 2011) and have a large SOC storage potential (Aerts et al., 2003; Sui and Zhou, 2013), whereof >95 % of the total C is stored as SOC (Grace, 2004). Moreover, it is likely that the response of these typically N limited systems (LeBauer and Treseder, 2008) to long-term N fertilization will be more pronounced than more southern grassland sites with higher background N deposition

(Hopkins et al., 2009). However, the role of chronic N inputs in the net SOC storage in these ecosystems is yet unclear.

In this study we therefore investigated the effect of chronically elevated N inputs on the northern C sink by quantifying the total ecosystem C stocks and net SOC storage rates of unmanaged Icelandic grasslands. We studied two site pairs with contrasting natural N inputs and soil developmental stage on the volcanic Vestmannaeyjar archipelago (south Iceland, Fig. 1). Variations in natural N inputs were caused by the topographical preferences of seabirds to form breeding colonies at

specific locations. As N is by far the most limiting element in the ecosystems under investigation (Aerts and Chapin, 2000; Leblans et al., 2014), the influence of other seabird derived nutrient inputs was assumed to be negligible. We distinguished between decadal (topsoil; 40–50 years) and millennial (total soil profile; 1,600 years) responses of net SOC storage to chronic N inputs by soil layer dating. Further, we focused on the importance of soil developmental stage, comparing early developmental (E) versus mature (M) soils under chronically low ($E_{NL}$ and $M_{NL}$) and high ($E_{NH}$ and $M_{NH}$) N input conditions.

We expected that soil N stocks and N availability would be smaller at sites with low natural N input rates than at sites with high natural N inputs, but we also expected that the early developmental stage with high N input rates ($E_{NH}$) would not have reached the total N stock and availability of the mature soils with low N inputs ($M_{NL}$). Hence, we expected the following sequence of soil N stocks and availability: $E_{NL} < E_{NH} < M_{NL} < M_{NH}$. Further we hypothesized that N availability would better explain soil C inputs (plant production) than soil N stocks in these N limited systems, as a large part of the N in andosols is

generally inaccessible for plant roots (Gudmundsson et al., 2004).

We hypothesized that the decadal net SOC storage rate of mature Icelandic grasslands would be stimulated by chronically elevated N inputs. We expected that this stimulation would persist at the millennial timescale, but that the effect would be smaller, as a consequence of SOC saturation processes in the subsoil (Zehetner, 2010; Olson and Al-Kaisi, 2015). Early developmental Icelandic grasslands with low natural N inputs were expected to have lower decadal net SOC storage rates

compared to mature grasslands because of their relatively higher N limitation. We did expect, for the same reason, that the relative positive effect of chronic N inputs would be more pronounced in the early developmental grassland sites. These hypotheses result in the following order of net SOC storage at the decadal timescale $E_{NL} <<< E_{NH} < M_{NL} < M_{NH}$ and at the millennial timescale $M_{NL} < M_{NH}$.

## 2 Material and methods

### 2.1 Study sites

This study was performed on three islands of the volcanic Vestmannaeyjar archipelago (63°25' N, 20°17' W; south Iceland; Fig. 1) in mid-July 2012 and 2013. The climate is cold temperate, with a mean annual temperature at Stórhöfði



(meteorological station on the main island, Heimaey) of 5.1 °C between 1963 and 2012, and a min. and max. monthly average of 1.3 and 9.6 °C. Mean annual precipitation during the same period was 1600 mm (Icelandic Meteorological Office). The main vegetation type on the Vestmannaeyjar archipelago are lush grasslands, except in areas that are unsuitable for seabird colonization where heathlands, herb slopes or dry meadows can be found (Magnússon et al., 2014).

Two pairs of sites with low and high natural N inputs (negligible and major seabird influence respectively) were established on islands where soils were either at an early developmental stage (E) or on islands with mature soils (M) (Magnússon et al., 2014). The low N sites ($E_{NL}$ and $M_{NL}$) received on average 1.3–1.4 kg N ha$^{-1}$ yr$^{-1}$ in the form of natural background N input by atmospheric deposition (Sigurdsson and Magnússon, 2010). No symbiotic $N_2$ fixing vascular plant species were found in any of the study plots (Magnússon et al., 2014). The high N input sites ($E_{NH}$ and $M_{NH}$) received on average 47 kg N ha$^{-1}$ yr$^{-1}$

in the case of $E_{NH}$ (Leblans et al., 2014) and 67 kg N ha$^{-1}$ yr$^{-1}$ in the case of $M_{NH}$, an estimation based on a bioenergetics model of Wilson et al. (2004) and Blackall et al. (2007) in combination with nesting densities from Hansen et al. (2011). All sites had similar bedrock characteristics (see further) and were located within 25 km distance from each other so that the influence of climate could be assumed to be negligible.

The $E_{NL}$ and $E_{NH}$ sites were located on the island Surtsey (Fig. 1), a 50-year-old volcanic island that was formed in an

eruption between 1963 and 1967. Both $E_{NL}$ and $E_{NH}$ were located on the lower plain of basaltic lava flows that are partly filled with sand and silt (Jakobsson et al., 2007). While $E_{NL}$ was virtually free of seabird influence, $E_{NH}$ was located inside the confines of a well-defined permanent breeding colony of lesser black backed seagulls (*Larus fuscus*), great black-backed gulls (*Larus matitiumus*) and herring gulls (*Larus argentatus*) that was established in 1986 on the southwestern part of the island (Leblans et al., 2014; Magnússon et al., 2014). Limited soil formation had taken place at $E_{NL}$, while the $E_{NH}$ soil

profile consisted of an O horizon, on top of a premature A horizon (max. 10 cm deep) and was classified as an Andosol (Arnalds, 2015). The pH at $E_{NL}$ was significantly higher than at $E_{NH}$ (7.6 vs. 6.6; Sigurdsson and Magnússon, 2010). The plant community at $E_{NL}$ was in an early successional transitional state between barrens and grassland, while the plant community at $E_{NH}$ had reached an early successional grassland stage (Magnússon et al., 2014).

The $M_{NL}$ site was located on Heimaey, the largest island of the Vestmannaeyjar archipelago (13.4 km$^2$) (Fig. 1). It was

established in Lyngfellisdalur, a valley on the southeastern part of the island. The valley is visually isolated from the sea, which makes it an unsuitable breeding location for seabirds. No seabird colonies were found within the valley and it is highly unlikely that they ever existed in the past because of the topographical conditions. The surfacing basaltic bedrock dates back to 5,900 AD (Mattsson and Hoskuldsson, 2005) and is covered by a mature soil classified as 'Brown Andosol' (Arnalds, 2008), which typically have a pH between 5.5 and 7.5 (Arnalds, 2015). The $M_{NL}$ site hosts a species-rich grassland

community, typical for low nutrient conditions (Magnússon et al., 2014). The $M_{NH}$ site was located on the nearby island Ellidaey (0.46 km$^2$) (Fig. 1) which hosts the second largest puffin colony (*Fratercula arctica*) of the archipelago, with 16,400 breeding pairs (Hansen et al., 2011). Due to its topographical conditions it is highly likely that the island has served as breeding ground for seabirds from early times. The $M_{NH}$ site has similarly aged bedrock and soil characteristics as the



nearby $M_{NL}$ site on Heimaey (Mattsson and Höskuldsson, 2003; Magnússon et al., 2014), but the nutrient-rich conditions have given rise to the development of a species-poor grassland community (Magnússon et al., 2014).

The soils of $M_{NL}$ and $M_{NH}$ contained two well-defined volcanic ash layers that could be used to date the profile; the lower one from a volcanic eruption in ca. 395 AD, which most probably originated from the mainland volcano Katla (Larsen, 1984), while the upper one originated from an eruption on Heimaey in 1973 (Morgan, 2000). Both ash layers varied in thickness between 0.5 and 5 cm. The 395 AD layer was located at $110 \pm 5$ (SE) cm soil depth at $M_{NL}$ and at $160 \pm 5$ (SE) cm soil depth at $M_{NH}$, and coincided with the maximum depth of undisturbed soil, as below it an eroded gravel layer was found. The 1973 AD layer was located at $6.4 \pm 0.4$ (SE) and $11.4 \pm 1.7$ (SE) cm soil depth at $M_{NL}$ and $M_{NH}$, respectively, and could be considered as the separation between topsoil and subsoil. At both sites, the topsoil contained over 70 % of the roots. At $E_{NL}$ and $E_{NH}$, the vegetation and soil development was too recent to detect the 1973 AD ash layer.

## 2.2 Experimental setup

At $E_{NL}$ and $E_{NH}$, our measurements were performed at ten and eight permanent 10x10 m research plots, respectively, that were established on Surtsey between 1990 and 1995 (Magnússon et al., 2014). Adjacent to each permanent plot, three 0.2x0.5 m subplots were placed for destructive measurements. In both $M_{NL}$ and $M_{NH}$, three 0.2x0.5 m subplots were placed adjacent to four 10x10 m research plots (n=4) that were established in 2013. The subplots at $M_{NL}$ and $M_{NH}$ were protected against possible human and livestock influence prior to the measurements (early May – late July) by covering them with 1x1 m enclosure cages. No such protection was needed for $E_{NL}$ and $E_{NH}$, since neither tourists nor domestic animals are permitted on the Surtsey island (Baldurson and Ingadóttir, 2007).

## 2.3 N availability

A relative measure for N availability was obtained using cation- and anion-exchange membranes (PRS™ probes, Western Ag Innovations Inc.; Saskatoon, SK, Canada). The membranes continuously absorb charged ionic species over the burial period, and the N availability is calculated as soil N flux over time. Four sets of membranes were inserted for one week in the topsoil (0–10 cm depth) of each main study plot in mid-July 2013. Afterwards, they were sent to Western Ag Innovations Inc. (Saskatoon, SK, Canada) for further analyses.

## 2.4 Plant analyses

All aboveground parts of vascular plants were harvested in each 0.2x0.5 m subplot, while litter and moss were collected in a 0.2x0.2 m section of the subplot. Subsequently, all vegetation samples were dried for 48 h at 40°C or until weight loss stopped, weighed, and milled using a ball mill (Retsch MM301 Mixer Mill, Haan, Germany) in preparation for further C and N analyses by dry combustion (Macro Elemental Analyser, model vario MAX CN, Hanau, Germany).





In each of the permanent plots, the height of three individuals of *Cerastium fontanum* was measured, as it was the only plant species found in all four treatments. Further, 2 g dry weight of mature healthy leaves were collected for analyses of N by dry combustion (NC2100 C/N analyser; Carlo Erba Instruments, Italy) and of P by inductively coupled plasma procedure (sequential ICP-OES spectrometer; Jobin Yvon Ultima 2, France), respectively.

**2.5 Soil analyses**

Underneath the vegetation sampling subplots (see Sect. 2.4), two parallel soil cores (8.67 cm diameter) were taken and split into 0–5, 5–10, 10–20 and 20–30 cm depth segments, where depth to the bedrock allowed this at $E_{NL}$ and $E_{NH}$. At $M_{NL}$ and $M_{NH}$, two additional 4.82 cm diameter soil cores were extracted down to the 395 AD ash layer from three out of the four main plots with a closed split corer and separated in segments of 30 cm. Each pair of soil cores was used to retrieve the dry

weight and C and N concentrations of the fine roots and of the soil fraction < 2 mm and to calculate the stoniness of the soil (the soil fraction > 2 mm; the C and N content of soil particles > 2 mm was assumed to be negligible). From the first of the two soil cores, the fine roots were washed out on a 0.5 mm sieve and subsequently treated identically to the aboveground vegetation. From the same core, the stoniness of the soil (fraction of soil particles > 2 mm) was derived using a sieve with a mesh size of 2 mm. The second soil core was dried for 48h at 40°C or until weight loss stopped, and the dry weight of soil <

2 mm was calculated using the following equation:

$$DW\ s_{<2mm} = DW\ s_{total} - DW\ s_{>2mm} - DW\ r \tag{1}$$

Where $DW$ is dry weight, $s_{<2mm}$ is the portion of soil particles < 2 mm, $s_{total}$ is total soil core, $s_{>2mm}$ is the portion of soil particles > 2 mm and $r$ are the fine roots.

Afterwards 2 g of soil < 2 mm was sieved from the second soil core and milled with a ball mill (Retsch MM301 Mixer Mill,

Haan, Germany) as preparation for further C and N analyses.

**2.6 Calculation of  C and N stocks and net SOC storage rates**

The C and N stocks of vegetation, roots and the soil fraction < 2 mm were calculated by multiplying the respective C and N concentration with the dry weight of the sample and correcting for the respective sampling size and depth.

The decadal net SOC storage rate (ton SOC ha$^{-1}$ yr$^{-1}$) was calculated by dividing the topsoil SOC stocks by their respective

accumulation time in years. Topsoil SOC stocks for $M_{NL}$ and $M_{NH}$ were those found above the 1973 ash layer (40 years of SOC storage). For $E_{NL}$, the maximum accumulation time was only 50 years, as the island Surtsey was formed in 1963. For $E_{NH,}$ the time of the initial seabird colonization of each plot was known (1986-2012; Magnússon et al., 2014) and the amount of SOC accumulated before the time of colonization (derived from the stocks in the $E_{NL}$ site, assuming a constant accumulation) was subtracted from the total stock prior to the calculation. Subsequently, the millennial net SOC storage rate

at $M_{NL}$ and $M_{NH}$ was calculated for consecutive cumulative soil ages, with 200 years intervals, down to the 395 AD ash



layer. The age of the soil was calculated by assuming a constant soil accumulation rate between the two ash layers (0.67 and 0.97 mm yr$^{-1}$ at $M_{NL}$ and $M_{NH}$, respectively).

## 2.7 Data analyses

The effects of the chronically elevated N inputs and soil developmental stage on N stocks, N availability, aboveground biomass, plant N/P ratios, plant height, ecosystem C stocks and net SOC storage rate in the topsoil, were tested with a two-way ANOVA, with N input (low/high) and soil developmental stage (early/mature) as fixed factors. In case of significant interaction, the pairwise differences were tested by post hoc LSD tests or Wilcoxon signed rank tests when the requirements of normality and homoscedasticity were not met. The change in net SOC storage rate with increasing cumulative soil age was tested for $M_{NL}$ and $M_{NH}$ with a two-way ANOVA, with N input (low/high) and cumulative soil age as fixed variables. The correlation between net SOC and net SON storage rate was tested with a Pearson correlation test (conditions of normality and homoscedasticity were met). All tests were performed in R software (R-core-team, 2014) and null hypotheses were rejected at $p < 0.05$.

## 3 Results

### 3.1 N availability and N limitation

The PRS-derived N availability in the main rooting zone (0–10 cm) was significantly lower at the low N input sites ($E_{NL}$ and $M_{NL}$) than at the high N input sites ($E_{NH}$ and $M_{NH}$) (Fig. 2.A; Table 1). Accordingly, plot-scale biomass, the height of *Cerastium fontanum* and leaf N/P ratios were significantly higher for the high N input sites (Fig. 3; Table 2). Nonetheless, leaf stoichiometry at the high N input sites still indicated N limitation (Fig. 3.A).

Soil developmental stage had a marginally significant effect on topsoil N availability, with the M having slightly higher N availability than the E (Fig. 2.A; Table 1). Plant height and aboveground biomass were significantly higher in M than in E, but leaf stoichiometry did not reveal a significant influence of soil developmental stage (Fig. 3; Table 2).

### 3.2 N stocks

The effect of chronically elevated N inputs on total N stocks followed contrasting patterns when only topsoil was taken into account or when the total soil profile above the 395 AD ash layer was included. In the topsoil, elevated N inputs increased the total ecosystem N stocks significantly, from 34 to 1080 kg ha$^{-1}$ for $E_{NL}$ and $E_{NH}$ and from 870 to 2200 kg ha$^{-1}$ for $M_{NL}$ and $M_{NH}$. This N input effect was significant for both plant biomass and soil stocks (Fig. 2.B; Table 1). When the total soil profile was considered (only applicable to $M_{NL}$ and $M_{NH}$), chronically elevated N inputs significantly increased N stock by 38 %, compared to 121 % in the topsoil (Fig. 2.C; Table 1). This difference was, however, only significant when the N

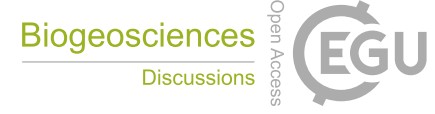

stocks were compared per cumulative soil age (similar to the analysis of the net C storage rate Sect. 2.6 and 2.7; data not shown), and not when the N stock in the bulk soil was considered as a whole (Table 1).

### 3.3 Effect of chronically elevated N inputs on C stocks and storage in biomass and topsoil

At the sites with mature soils ($M_{NL}$ and $M_{NH}$), chronically elevated N inputs led to a doubling of the ecosystem biomass plus
topsoil C stock, from 17 to 35 ton C ha$^{-1}$ (Fig. 4.A; Table 3). This significant effect of chronically elevated N input on the total C stocks was also visible in almost all individual C stocks: the SOC stock doubled from approximately 10 to about 20 ton C ha$^{-1}$ and the C stock in aboveground vascular plant biomass ("shoots") increased almost 4 times, from 1.0 to 3.4 ton C ha$^{-1}$ (Fig. 4.A; Table 3). The increase in root C stock in the topsoil of $M_{NH}$ compared to $M_{NL}$ was only marginally significant, but the total root C stock, including deeper roots, was significantly higher at $M_{NH}$ (Fig. 4.A and 4.B; Table 3). The litter C
stock remained stable at approximately 1 ton ha$^{-1}$ and the moss C stock decreased from 1 to 0.2 ton ha$^{-1}$ between $M_{NL}$ and $M_{NH}$ (Fig. 4.A, Table 3). The net SOC storage rate in the topsoil of the sites with mature soils was 50 % higher under chronically increased N inputs (from 0.30 to 0.44 ton C ha$^{-1}$ yr$^{-1}$; Fig. 5.B; Table 4).

For the sites in early soil developmental stage ($E_{NL}$ and $E_{NH}$), the effects of chronically elevated N input were even more pronounced, with all C stocks in biomass and soil significantly higher in $E_{NH}$ than in $E_{NL}$ (Fig. 4.A, Table 3). Biomass C
stocks increased from 0.2 to 15.0 ton ha$^{-1}$, total ecosystem C stock increased by from 1.0 to 22.1 ton ha$^{-1}$ and SOC storage in topsoil increased from 0.02 to 0.28 ton C ha$^{-1}$ yr$^{-1}$.

Regarding the effect of developmental stage, comparison of the early developmental and mature sites over the same timeframe (the last 40 to 50 years) revealed that the total C stock of $E_{NL}$ was only a fraction of $M_{NL}$ (1.0 vs. 30 ton ha$^{-1}$; Fig. 4.A; Table 3), and that its net SOC storage rate was a mere 9 % of the rate in $M_{NL}$ (Fig. 5.B; Table 4). At the high N inputs
sites, on the contrary, all biomass C stocks of $E_{NH}$, had reached the same level as $M_{NH}$. This was not the case for the SOC stocks, which, nevertheless, reached half of the $M_{NH}$ stock (Fig. 4.A; Table 3). In terms of net SOC storage rate, $E_{NH}$ stored half as much SOC as $M_{NH}$ per unit time, and had reached the same rate as $M_{NL}$ (Fig. 5.B; Table 4).

### 3.4 SOC stocks and storage in the total soil profile

The SOC stocks and storage rates in the total soil profile (since 395 AD) could only be studied at the sites with mature soils
($M_{NL}$ and $M_{NH}$). At both sites, the C stocks and C storage rate decreased significantly with increasing soil depth (Fig. 5; Table 4). However, in ca. 1000 years old soil layers, the C storage reached an equilibrium of 0.12 and 0.16 ton C ha$^{-1}$ yr$^{-1}$ for $M_{NL}$ and $M_{NH}$ respectively. These results also indicate that the chronically elevated N inputs caused a significant increase in net SOC storage of $M_{NH}$ compared to $M_{NL}$, and that this was maintained throughout the soil profile down to the 395 AD ash layer (Fig. 5.B; Table 4). The effect size of chronically elevated N inputs on net C storage rate decreased with cumulative
soil age, as was shown by the significant interaction between N inputs and cumulative soil age (Table 4), until an equilibrium of ca. + 0.04 ton SOC ha$^{-1}$ yr$^{-1}$ (equivalent to +25 %) was reached. When the SOC stocks of $M_{NL}$ and $M_{NH}$ were compared as



simple bulk values, integrating the entire soil profile, the differences between $M_{NL}$ and $M_{NH}$ were not statistically significant (Fig. 4.B; Table 3).

Finally, taking the whole measurement depth into account (30 cm for $E_{NL}$ and $E_{NH}$ and up to the 395 AD ash layer for $M_{NL}$ and $M_{NH}$), the net storage rates of SOC and SON were strongly linked over all treatments, with an average C/N ratio of 12 (Fig. 6).

## 4 Discussion

### 4.1 N availability and N stocks in unmanaged Icelandic grasslands

The N availability and the total SON stocks were greatly increased in the high N input sites compared to the low N input sites. Interestingly, even though this higher N availability and N stocks at the high N input sites clearly stimulated biomass production and C storage (biomass stocks, plant height and ecosystem C stocks were significantly higher), the plant N/P ratios indicated that plant growth remained N limited, in spite of the relatively high chronic N input rates in both $E_{NH}$ and $M_{NH}$ (~47 and ~67 kg N ha$^{-1}$ yr$^{-1}$, respectively).

Contrary to our expectations that millennia of soil development would increase N availability and N stocks more than a few decades of allochtonous N inputs, the N status was clearly more closely related to the annual seabird-derived N input than to ecosystem maturation. Thousands of years of N retention and recycling did (partly) alleviate the N-limitation in $M_{NL}$ and $M_{NH}$ compared to $E_{NL}$ and $E_{NH}$ respectively, as was shown by the significantly higher standing biomass and plant height. However, the N availability was only slightly higher in mature sites compared to the early developmental sites. The N stock, on the other hand, was indeed increased significantly in $M_{NL}$ compared to $E_{NL}$. However, the observation that $M_{NL}$ was still greatly N-limited (indicated by the low plant N/P stoichiometry), supported our presumption that most of the accumulated ecosystem N stock was locked up in undecomposed soil organic matter or in biomass. In the high N input sites, the short timeframe in which $E_{NH}$ had received increased N inputs had been sufficient to bring the vegetation N stocks of $E_{NH}$ and $M_{NH}$ at the same level. The SON stocks, however, were still significantly higher in $M_{NH}$, which shows that the SON stock were not yet in equilibration with the N input rates in $E_{NH}$.

We expected that N availability would better explain differences in N limitation and biomass production than N stocks, because of the typical strong binding of N to soil particles in volcanic soils (Gudmundsson et al., 2004). Nonetheless, both had a comparable explanatory value.

### 4.2 Net SOC storage in mature Icelandic grasslands under chronically elevated N inputs

### 4.2.1 Decadal time scale

Our study was conducted on Andosols, which have specific characteristics, including high concentrations of Al, Fe and Si that can e.g. bind SOM in 'metal-humus' complexes (Arnalds, 2015). Nonetheless, the decadal net SOC storage rate in the



mature grasslands (0.30 and 0.44 ton ha$^{-1}$ yr$^{-1}$ at M$_{NL}$ and M$_{NH}$ respectively) corresponded well with the average topsoil SOC storage in a broad range of soil types under previously SOC depleted perennial temperate grasslands (0.33 ton ha$^{-1}$ y$^{-1}$; Post and Kwon, 2000).

As hypothesized, chronically elevated N inputs stimulated decadal SOC storage in mature soils, but to a lesser extent than in early developmental soils (an increase of 50 % from MNL to MNH, compared to an increase of 250 % from ENL to ENH; Leblans et al., 2014). The magnitude of the response is in line with an earlier long-term N addition study on managed grasslands in east Iceland, which reported a > 50 % increase in SOC concentration in the upper 10 cm of the soil after 43 years of fertilization by 120 kg N ha$^{-1}$ yr$^{-1}$ (Gudmundsson et al., 2004). The strong positive effect of chronically elevated N inputs on decadal net SOC storage in our study is in line with the theory that the recent northern C sink is at least partly caused by increasing N deposition (Hudson et al., 1994; Lloyd, 1999; Schlesinger, 2009).

The net response of decadal SOC storage to chronically elevated N inputs depends on the responses of the individual processes that influence SOC uptake and release in the topsoil: (1) net primary productivity (NPP), (2) C partitioning and (3) decomposition rate (Mack et al., 2004; Batjes, 2014). (1) Chronically elevated N inputs greatly stimulated NPP in the present study, which agreed with what is generally found in studies that investigate the effects of N inputs on productivity in northern grasslands (Sillen and Dieleman, 2012). (2) The root/shoot C partitioning was similar for M$_{NL}$ and M$_{NH}$ (average root/shoot ratios of ~10; data not shown). However, since the total amount of belowground C allocation is potentially also affected by changes in root turnover rates (Aerts et al., 1992; Milchunas et al., 2005), changes in exudation and mycorrhizal activity (Vicca et al., 2012) and changes in microbial C use efficiency (Wieder et al., 2013), the total belowground C inputs cannot be derived from the present data. (3) The response of decomposition to chronically elevated N inputs was not assessed in the present study, but previous studies have reported both positive and negative N input effects on the decomposition rate in the topsoil of northern grasslands. The direction of the response depended on natural background N deposition, N input rate and litter quality (Aerts et al., 2003; Knorr et al., 2005; Hobbie, 2008; Zhang et al., 2008). In any case, the increased SOC stocks clearly indicate that the increase in topsoil C input rate surpassed potential increases in decomposition rate in our study.

**4.2.2 Millennial time scale**

The total SOC stocks (down to the 395 AD ash layer; 220–280 ton C ha$^{-1}$) in the present study did not only correspond closely to a previous estimation for Brown Andosols in Iceland (227 ton ha$^{-1}$; Óskarsson et al., 2004), but were also in line with non-volcanic temperate grassland soils, where estimates range from 197 (Schlesinger, 1997) to 236 ton ha$^{-1}$ (Janzen, 2004). The observed millennial net SOC storage rates (0.12–0.16 ton C ha$^{-1}$ yr$^{-1}$) corresponded well with those of deep SOC rich soils in northern regions (0.15–0.30 ton ha$^{-1}$ yr$^{-1}$; Trumbore and Harden, 1997) and with the long-term SOC storage in temperate grassland ecosystems in China (0.11 ton ha$^{-1}$ y$^{-1}$; He and Tang, 2008).



As hypothesized, the millennial net SOC storage rate was much lower than the decadal storage rate. This agrees with a recent review study of Matus et al. (2014) that showed a general decrease in net SOC storage rate with depth in Andosols and with the general observation that net SOC storage rates decline when soils approach their mature state (Post and Kwon, 2000). However, there was no evidence that the mature sites in this study had reached an SOC steady state, as the decline in net

SOC storage rate with depth (or with increasing cumulative soil age) stabilized around 1000 years before present and did not decline to zero.

The stable C/N ratio of about 12 in the total soil profile of both $M_{NL}$ and $M_{NH}$ suggested that the total SOC stock could continue to increase with elevating N inputs even after millennia of soil maturation, providing that N can be retained. This was supported by our observation that millennial net SOC storage rate was still increased under chronically elevated N

inputs, albeit to a lesser extent than the decadal storage rate (25 % vs. 50 % increase, respectively). The modest increase in SOC storage rate under chronically elevated N inputs, however, was consistent throughout the soil profile and added up to a considerable strengthening of the SOC sink over a long time span while a thicker soil was developed.

### 4.3 Importance of soil developmental stage for net SOC storage in Icelandic grasslands

As expected, the decadal net SOC storage rate in mature soils under low natural N inputs was substantially higher compared

to early developmental soils under low natural N inputs. The observed increase in SOC storage (~0.30 ton ha$^{-1}$ yr$^{-1}$) was within the range of estimates for the transformation from early developmental to mature Andosols which range from 0.1 ton ha$^{-1}$ yr$^{-1}$ (Vilmundardóttir et al., 2015). This might be an underestimation as the older soils in this study had not reached their mature state yet, but mature soils in Iceland have been found to have a SOC storage rate of 0.6 ton ha$^{-1}$ yr$^{-1}$, on average (Óskarsson et al., 2004). The maturation-driven increase in net SOC storage rate is not only an Andosol feature, but is also

generally found in other soil orders (e. g. Lichter, 1998; Foote and Grogan, 2010; Kabala and Zapart, 2012; Kalinina et al., 2013) and is caused by centuries of N-accumulation, the stimulation of internal N cycling through biomass and the gradual increase in SOC stock (Vitousek and Reiners, 1975; Kirschbaum et al., 2003). Increasing SOC improves the N exchange capacity and the water holding capacity of the soil (Deluca and Boisvenue, 2012), thus stimulating plant growth and net SOC storage.

We did not expect that only 28 years of elevated N inputs at the $E_{NH}$ site would have created a positive effect on decadal net SOC storage rate similar to that reached at $M_{NL}$ after thousands of years of slow N retention and recycling. In fact, the current annual SOC storage rate of $E_{NH}$ most likely surpassed $M_{NL}$, as it was probably higher in recent years than its average rate calculated since the start of seabird colonization. Indeed, $E_{NH}$ showed a 10-years delay in the establishment of full surface cover (and consequently net SOC storage) after the initiation of the allochthonous N inputs (Magnússon and

Magnússon, 2000). Hence, $E_{NH}$ was likely approaching the decadal net SOC storage rate of $M_{NH}$. This supposition is supported by the similar biomass C stocks at $E_{NH}$ and $M_{NH}$, but it can be expected that the development and stabilization of





soil processes that regulate net SOC storage will need longer time to come into equilibration with the C input rates (Post and Kwon, 2000; Creamer et al., 2011).

If we assume that, with time, the early developing soils will evolve into similar soils as the soils on the older islands, then the difference in total SOC stock provides an idea of the future 'SOC gap' (Kramer and Gleixner, 2008). This gap amounted to 220–260 ton SOC ha$^{-1}$ in the present study and will be gradually filled during the process of soil maturation by the input of organic material at the soil surface and into the rooting zone. Chronically elevated N inputs induced only a slight (20 %) increase in this SOC gap, but did have a large influence on the time frame in which the gap could be filled (by enhancing the net SOC storage rate at $E_{NH}$ by a factor 16). A key question in relation to this is when the apparent SOC gap will saturate. It has been shown that this depends on the status of the SOC stabilizing processes in the subsoil (Fontaine et al., 2007; Rumpel and Kogel-Knabner, 2011; Pausch and Kuzyakov, 2012) and on the frequency of soil disturbance processes (Baldocchi, 2008). Until recently, it was assumed that all mature soils were SOC-saturated (Wutzler and Reichstein, 2007). However, this supposition has been challenged by various observations of continuously increasing SOC stocks in old (> 1,000 years) undisturbed soils (e.g. Harden et al., 1992; Wardle et al., 1997) and model-based predictions that the equilibration process of SOC stocks could take millennia (Wang and Hsieh, 2002). Also our mature study sites ($M_{NL}$ and $M_{NH}$), which considered the past 1,600 years (since a large-scale disturbance before 395 AD), did not seem to have reached SOC saturation yet (see Sect. 4.2.2.).

Considering the large difference in C dynamics that was observed between early developmental and mature sites, it is important to bear in mind that many ecosystems are in an intermediate developmental stage, following past disturbances. Therefore, we stress the importance of taking soil developmental stage into account when estimating net SOC storage rates.

## 5 Conclusion

In our study, the decadal net SOC storage rate of mature Icelandic grasslands was greatly stimulated by chronically elevated N inputs, which supported the theory that the increasing northern terrestrial C sink during the past decades could be (partly) caused by increasing anthropogenic N inputs. The positive influence of chronic N inputs on the net SOC storage rate also persisted at a millennial timescale in the present study, albeit to a smaller extent. This indicates not only that mature Icelandic grasslands, if not disturbed, could remain C sinks if the current climate conditions prevail, but also that chronically elevated N inputs could induce a permanent strengthening of this sink.





**Author contribution**

B.D. Sigurdsson, B. Magnússon, N.I.W. Leblans and I.A. Janssens designed the study and N.I.W. Leblans and B.D. Sigurdsson carried it out. N.I.W. Leblans prepared the manuscript with contributions from all co-authors.

**Acknowledgments**

This research was supported by the Research Foundation – Flanders (FWO aspirant grant to NL; FWO postdoctoral fellowship to SV), the European Research Council Synergy grant 610028 (IMBALANCE-P), and the Research Council of the University of Antwerp. We acknowledge support from FSC-Sink, CAR-ES and the ClimMani COST Action E1308. The Surtsey Research Society, Institute of Natural History, Mogilsá – Icelandic Forest Research, Reykir and Keldnaholt – Agricultural University of Iceland and the Icelandic Coastguard provided logistical support for the present study. We thank
Sturla Fridriksson, Sigurdur Magnússon and Erling Ólafsson for generously providing access to their data. We are grateful to Anette Th. Meier for designing the map. We thank Annemie Vinck, Paul Leblans, Pieter Roefs, Rafaële Thuys, Elín Guðmundsdóttir, Elías Óskarsson, Sigurður Sturla Bjarnasson, Hekla Hrund Bjarnadóttir, Alexander Meire, Damiano Cillio, Linde Leblans and Dries De Pauw for their helping hands in the field. Further, we thank Brita Berglund, Baldur Vigfusson, Nadine Calluy, Marijke Van den Bruel and Els Oosterbos for their assistance with the lab analyses.

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

**Figure 1: Topographical map of the research area. (A) Location of the Vestmannaeyjar including the three study islands, Surtsey, Heimaey and Ellidaey at the southwest coast of Iceland. (B) and (C) show the islands Ellidaey and Surstey in more detail. The dotted outline on panel C shows the contours of the seabird colony (that was established anno 1986) in 2012. Dots show the research plots at early soil developmental stage under low (○ = $E_{NL}$) and high (● = $E_{NH}$) seabird-derived N inputs. Triangles show the research plots at sites with mature soils under low (△ = $M_{NL}$) and high (▲ = $M_{NH}$) seabird-derived N inputs respectively. Map designed by Anette Th. Meier.**



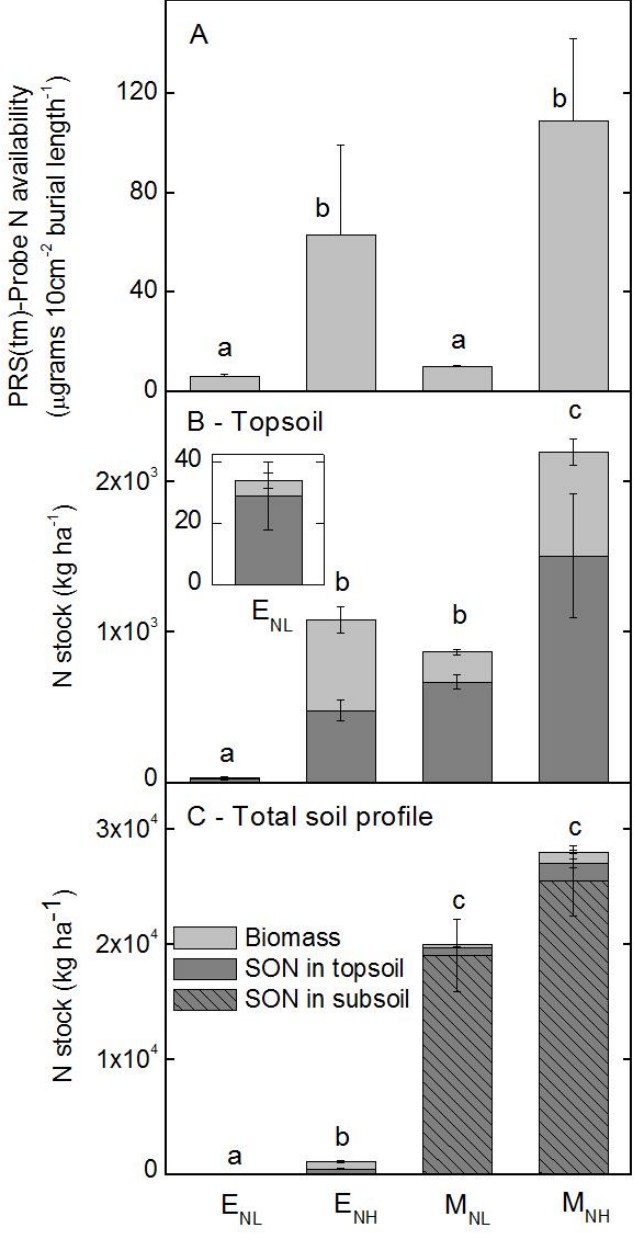

**Figure 2: (A)** PRS-probe derived N availability, measured by cation- and anion-exchange membranes that continuously absorb charged ionic species over the burial period, and expressed as soil N flux over time. **(B and C)** Nitrogen stocks in biomass (shoots + roots) and soil organic nitrogen (SON) in unmanaged Icelandic grasslands under low and high N inputs at an early soil developmental stage ($E_{NL}$ and $E_{NH}$) and at sites with mature soils ($M_{NL}$ and $M_{NH}$). The SON stocks are shown separately for the topsoil (B; since 1963 for $E_{NL}$ and $E_{NH}$, above the 1973 ash layer for $M_{NL}$ and $M_{NH}$) and the total soil profile including the subsoil (C; above the 395 AD ash layer). The inserted graph shows the N stock at $E_{NL}$ in detail and is valid for both panels B and C, as $E_{NL}$ had no subsoil. Letters show significant differences in total ecosystem N stocks. Error bars indicate SE's. Statistical details can be found in Table 1.





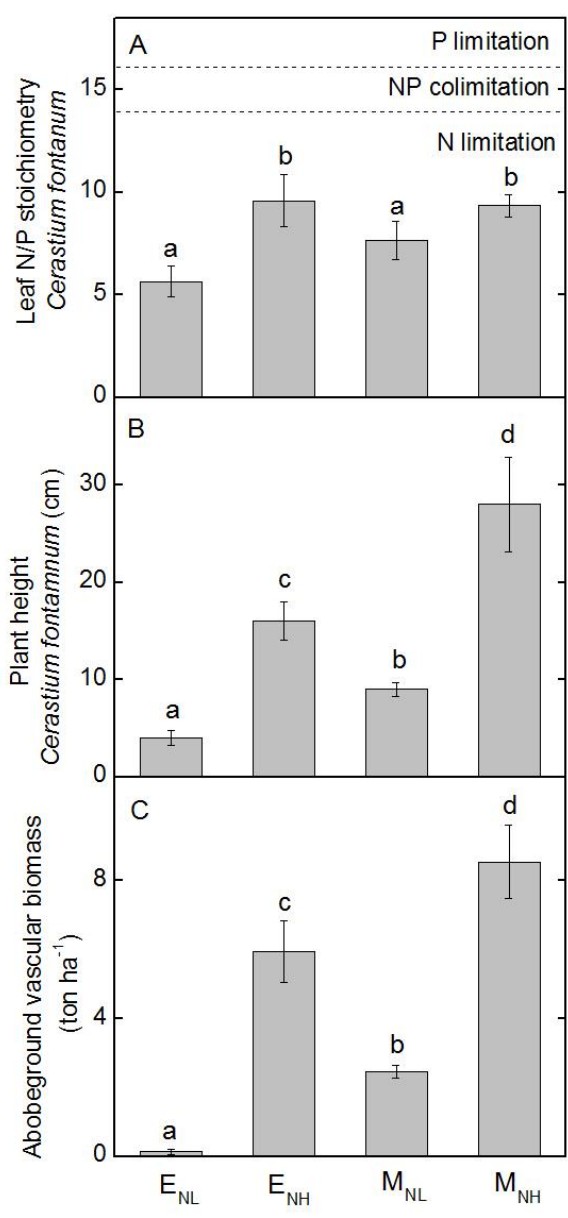

**Figure 3: (A) Leaf N/P stoichiometry of *Cerastium fontanum* in mature healthy leaves. Dotted lines show the borders of N limitation (N/P < 14), NP co-limitation (14 < N/P < 16) and P limitation (N/P > 16) for higher plant communities (Aerts and Chapin, 2000). (B) Plant height of *C. fontanum*. (C) Aboveground vascular biomass (monocots and dicots). Different bars shown unmanaged Icelandic grasslands under low and high N inputs at an early soil developmental stage (E$_{NL}$ and E$_{NH}$) and at sites with mature soils (M$_{NL}$ and M$_{NH}$). Error bars indicate SE's. Letters show significant differences. Further statistical details can be found in Table 2.**





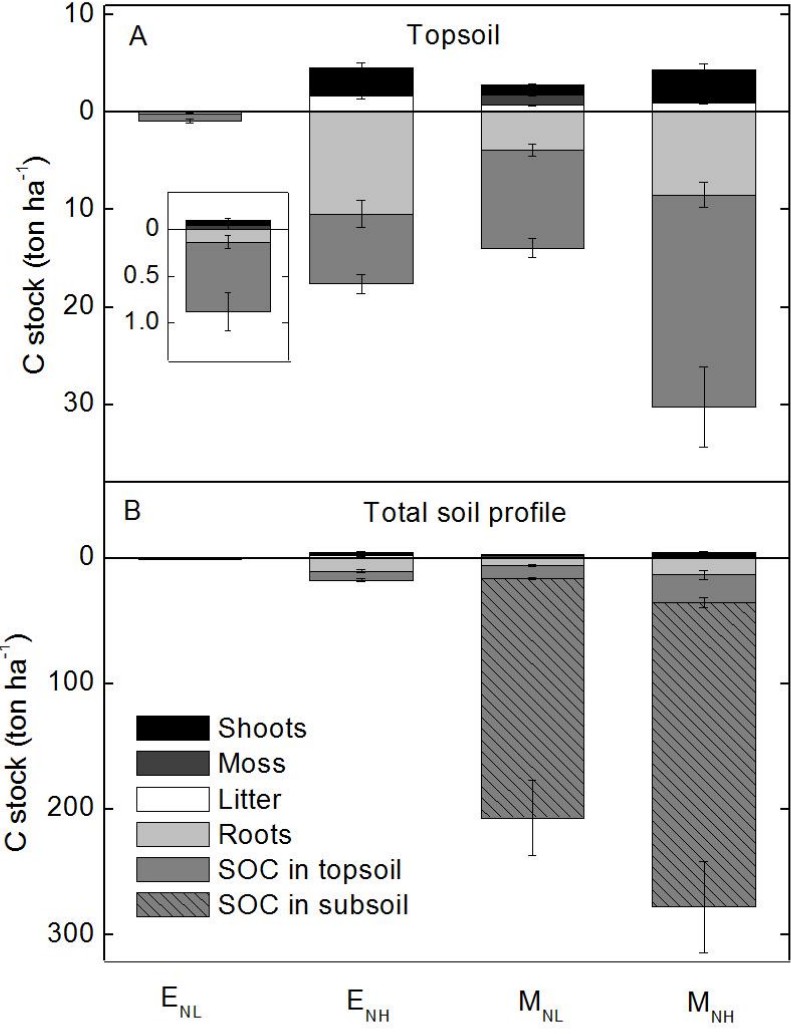

**Figure 4: Carbon stocks in biomass (shoots, moss, litter and roots) and soil organic carbon (SOC) in unmanaged Icelandic grasslands under low and high N inputs at an early soil developmental stage ($E_{NL}$ and $E_{NH}$) and at sites with mature soils ($M_{NL}$ and $M_{NH}$). The SOC stocks are shown separately for the topsoil (A; since 1963 for $E_{NL}$ and $E_{NH}$, above the 1973 ash layer for $M_{NL}$ and $M_{NH}$) and for the total soil profile including the subsoil (B; above the 395 AD ash layer). The inserted graph shows the C stock at $E_{NL}$ in detail and is valid for both panels as $E_{NL}$ had no subsoil. Error bars indicate SE's. Statistical details can be found in Table 3.**





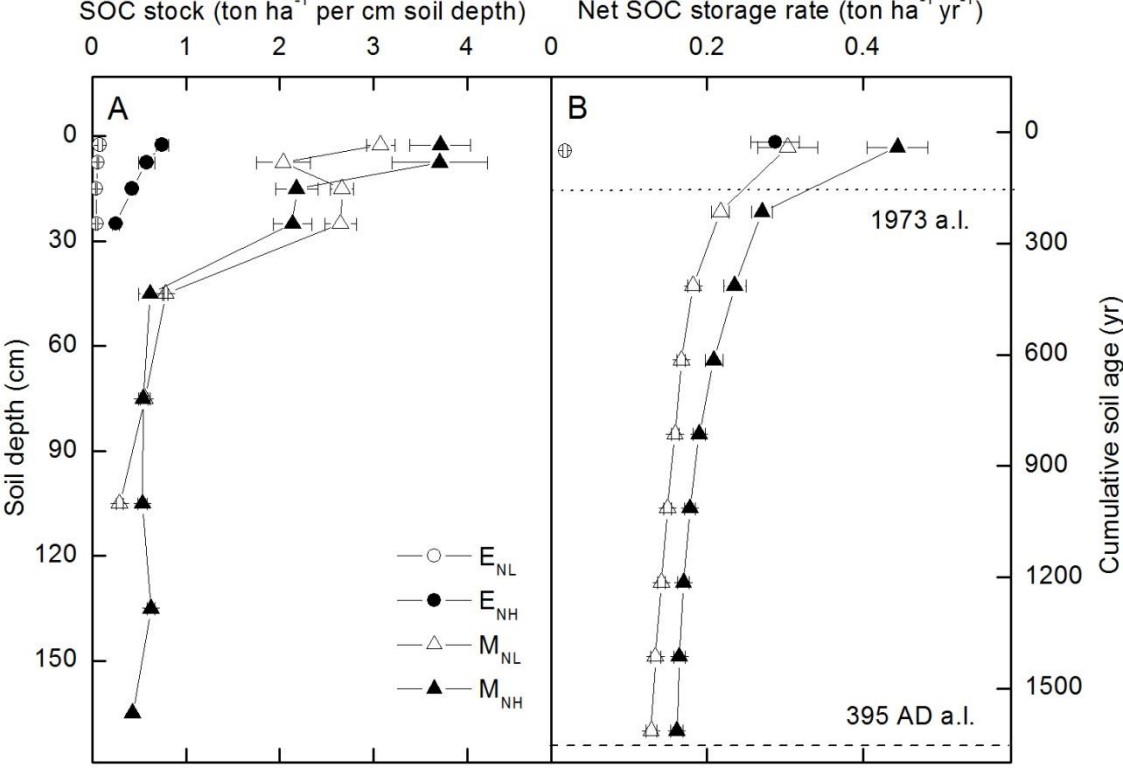

**Figure 5: Soil organic carbon (SOC) stocks and the derived net SOC storage rates in unmanaged Icelandic grasslands under low and high N inputs at an early soil developmental stage ($E_{NL}$ and $E_{NH}$) and at sites with mature soils ($M_{NL}$ and $M_{NH}$). (A) Depth profile of the SOC stocks per soil layer of one cm, derived from layers of 0–5, 5–10, 10–20, 20–30, 60–90, 90–120, 120–150 and 150– 180 cm soil depth. For $E_{NL}$ and $E_{NH}$, the SOC stocks were measured down to the bedrock or down to 30 cm soil depth, for $M_{NL}$ and $M_{NH}$ up to the 395 AD ash layer. Note the difference in depth of the 395 AD ash layer between $M_{NL}$ and $M_{NH}$. (B) Net SOC storage rate per cumulative soil age, where soil age was calculated assuming a linear soil accumulation between the 395 AD and 1973 ash layer. The dotted line indicates the 1973 ash layer, which marks the border between the topsoil and the subsoil. The dashed line indicates the 395 AD ash layer, which marks the lower limit of undisturbed soil. Note that $E_{NL}$ and $E_{NH}$ only contain topsoil, as they were still too young to have developed a deep organic soil. Error bars indicate SE's. Statistical details can be found in Tables 3 and 4.**





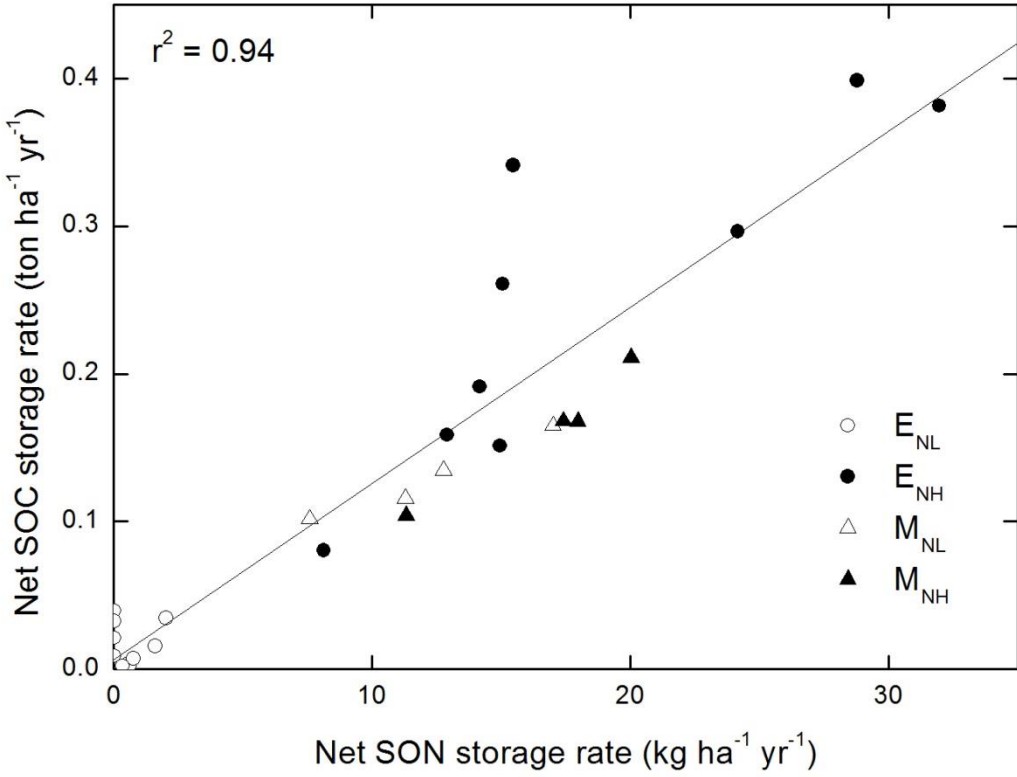

**Figure 6: Linear relationship between net soil organic carbon (SOC) storage rate and net SON storage rate for unmanaged Icelandic grasslands under low and high N inputs at an early soil developmental stage ($E_{NL}$ and $E_{NH}$) and at sites with mature soils ($M_{NL}$ and $M_{NH}$). The net SOC and SON storage rates were calculated over the whole soil profile (down to the bedrock or down to 30 cm for $E_{NL}$ and $E_{NH}$ and down to the 395 AD ash layer for $M_{NL}$ and $M_{NH}$). The linear relationship was highly significant ($p < 0.001$), with a slope of 0.12 ($\pm$ 0.02).**




**Table 1: Results of the two-way ANOVA's for N availability and the N stock in biomass, soil (soil organic nitrogen; SON) and the total ecosystem, using N input and soil developmental stage as fixed variables. The topsoil refers to the upper 30 cm or down to the bedrock for $E_{NL}$ and $E_{NH}$ (accumulated since 1963) and to the soil layer above the 1973 ash layer for $M_{NL}$ and $M_{NH}$. The total soil profile is only applicable to $M_{NL}$ and $M_{NH}$ and refers to the soil profile above the 395 AD ash layer. Significant source variables ($p < 0.05$) are indicated with an asterisk: \*p is 0.05–0.01, \*\*p is 0.01–0.001, \*\*\*p < 0.001.**

| | N availability | N stock (kg ha$^{-1}$) | | | | |
| | | Biomass | Topsoil | | Total soil profile[b] | |
| | | | SON | Total[c] | SON | Total[d] |
|---|---|---|---|---|---|---|
| **N input x soil developmental stage** | | | | | | |
| *Df numerator* | 1 | 1 | 1 | 1 | - | - |
| *Df denominator* | 21 | 23 | 23 | 23 | - | - |
| *F-value* | 0.47 | 0.58 | 2.03 | 0.74 | - | - |
| *p-value* | 0.50 | 0.45 | 0.17 | 0.40 | - | - |
| **N input** | | | | | | |
| *Df numerator* | na | 1 | 1 | 1 | 1 | 1 |
| *Df denominator* | na | 25 | 24 | 25 | 6 | 6 |
| *F-value* | 0[a] | 68.1 | 19.60 | 25.37 | 2.80 | 3.54 |
| *p-value* | \*\*\* | \*\*\* | \*\*\* | \*\*\* | 0.15 | 0.11 |
| **Soil developmental stage** | | | | | | |
| *Df numerator* | na | 1 | 1 | 1 | - | - |
| *Df denominator* | na | 24 | 24 | 24 | - | - |
| *F-value* | 101[a] | 4.69 | 21.05 | 38.47 | - | - |
| *p-value* | 0.06 | \* | \*\*\* | \*\*\* | - | - |

[a] Non parametrical Wilcoxon signed rank test: W-value

[b] Only applicable to mature soils ($M_{NL}$ and $M_{NH}$)

[c] Cumulation of biomass N and SON in topsoil

[d] Cumulation of biomass N and SON in total soil profile





**Table 2: Results of the two-way ANOVA's for leaf N/P stoichiometry of mature healthy leafs of *Cerastium fontanum*, plant height of *C. fontanum* and aboveground vascular plant biomass using N input and soil developmental stage as fixed variables. Significant source variables (p < 0.05) with an asterisk: \*p is 0.05–0.01, \*\*p is 0.01–0.001, \*\*\*p < 0.001.**

|  | Leaf N/P stoichiometry *C. fontanum* | Plant height *C. fontanum* | Aboveground vascular biomass |
|---|---|---|---|
| N input x soil developmental stage |  |  |  |
| *Df numerator* | 1 | 1 | 1 |
| *Df denominator* | 12 | 25 | 74 |
| *F-value* | 1.12 | 2.29 | 0.04 |
| *p-value* | 0.31 | 0.14 | 0.85 |
| N input |  |  |  |
| *Df numerator* | 1 | 1 | 1 |
| *Df denominator* | 14 | 27 | 76 |
| *F-value* | 7.16 | 33.29 | 75.23 |
| *p-value* | * | *** | *** |
| Soil developmental stage |  |  |  |
| *Df numerator* | 1 | 1 | 1 |
| *Df denominator* | 13 | 26 | 75 |
| *F-value* | 0.15 | 14.41 | 12.23 |
| *p-value* | 0.70 | *** | *** |



**Table 3: Results of the two-way ANOVA's for C stocks in different ecosystem parts, using N input and soil developmental stage as fixed variables. In case of significant interaction, no overall effects of N input or soil developmental stage could be derived and the pairwise differences were tested by post hoc LSD tests or Wilcoxon signed rank tests (lower part of the table). Topsoil C stocks corresponded to C stocks accumulated since 1963 for $E_{NL}$ and $E_{NH}$, and to C stocks that were accumulated since the deposition of the 1973 ash layer for $M_{NL}$ and $M_{NH}$. Total soil profile was only applicable to $M_{NL}$ and $M_{NH}$ ($E_{NL}$ and $E_{NH}$ have no subsoils yet), so that the influence of developmental stage could not be tested. It corresponded to the SOC stocks above the 395 AD ash layer. "Shoots" include all aboveground living vascular plant parts; SOC is Soil Organic Carbon. Significant source variables (p < 0.05) with respect to ecosystem parts are indicated with an asterisk: \*p is 0.05–0.01, \*\*p is 0.01–0.001, \*\*\*p < 0.001.**

| | | | | C stock (ton ha$^{-1}$) | | | | | |
| | | | | Topsoil | | | Total soil profile[a] | | |
| | Shoots | Moss | Litter | Roots | SOC | Total[b] | Roots | SOC | Total[c] |
|---|---|---|---|---|---|---|---|---|---|
| **N input x soil developmental stage** | | | | | | | | | |
| *Df numerator* | 1 | 1 | 1 | 1 | 1 | 1 | - | - | - |
| *Df denominator* | 23 | 23 | 22 | 23 | 23 | 23 | - | - | - |
| *F-value* | 0.06 | 6.58 | 7.08 | 6.58 | 3.17 | 0.34 | - | - | - |
| *p-value* | 0.81 | * | * | * | 0.08 | 0.56 | - | - | - |
| **N input** | | | | | | | | | |
| *Df numerator* | 1 | - | - | - | 1 | 1 | 1 | 1 | 1 |
| *Df denominator* | 25 | - | - | - | 25 | 25 | 6 | 6 | 6 |
| *F-value* | 33.30 | - | - | - | 10.15 | 8.57 | 11.38 | 0.68 | 2.42 |
| *p-value* | *** | - | - | - | ** | ** | * | 0.24 | 0.17 |
| **Soil developmental stage** | | | | | | | | | |
| *Df numerator* | Na | - | - | - | 1 | 1 | - | - | - |
| *Df denominator* | Na | - | - | - | 25 | 25 | - | - | - |
| *F-value* | 111[d] | - | - | - | 27.26 | 34.56 | - | - | - |
| *p-value* | 0.07 | - | - | - | *** | *** | - | - | - |
| **Effect of N input at early soil developmental stage** | | | | | | | | | |
| *Df numerator* | - | na | 1 | na | - | - | - | - | - |
| *Df denominator* | - | na | 16 | na | - | - | - | - | - |
| *F-value* | - | 35[d] | 24.24 | 0[d] | - | - | - | - | - |
| *p-value* | - | 0.28 | *** | *** | - | - | - | - | - |
| **Effect of N input at mature soils** | | | | | | | | | |
| *Df numerator* | - | 1 | 1 | na | - | - | - | - | - |
| *Df denominator* | - | 6 | 6 | na | - | - | - | - | - |
| *F-value* | - | 62.69 | 1.98 | 1[d] | - | - | - | - | - |
| *p-value* | - | *** | 0.21 | 0.06 | - | - | - | - | - |
| **Effect of soil developmental stage at low N input** | | | | | | | | | |
| *Df numerator* | - | na | 1 | na | - | - | - | - | - |
| *Df denominator* | - | na | 12 | na | - | - | - | - | - |
| *F-value* | - | 40[d] | 261.55 | 40[d] | - | - | - | - | - |
| *p-value* | - | ** | *** | ** | - | - | - | - | - |
| **Effect of soil developmental stage at high N input** | | | | | | | | | |
| *Df numerator* | - | na | 1 | 1 | - | - | - | - | - |
| *Df denominator* | - | na | 10 | 11 | - | - | - | - | - |
| *F-value* | - | 19[d] | 1.58 | 1.07 | - | - | - | - | - |
| *p-value* | - | 0.93 | 0.23 | 0.32 | - | - | - | - | - |

[a] Only applicable to mature soils ($M_{NL}$ and $M_{NH}$)

[b] Cumulation of biomass C and SOC in the topsoil

[c] Cumulation of biomass C and SOC in the total soil profile

[d] Non parametrical Wilcoxon signed rank test: W-value



**Table 4: Results of the two-way ANOVA's for the net soil organic carbon (SOC) storage rate in the topsoil (decadal net SOC storage rate) and the total soil profile (millennial net SOC storage rage). For the topsoil (left columns), N input and soil developmental stage were used as fixed factors. Topsoil corresponded to the net SOC storage rate since 1963 (down to 30 cm soil depth or down to the bedrock) in the case of $E_{NL}$ and $E_{NH}$, and to the net SOC storage rate in the soil above the 1973 ash layer for**
5 **$M_{NL}$ and $M_{NH}$. For the total soil profile (right columns; only applicable to $M_{NL}$ and $M_{NH}$), N input and cumulative soil age were used as fixed factors. Total soil profile corresponded to the soil above the 395 AD ash layer. Significant source variables (p < 0.05) are indicated with an asterisk: ns. is p > 0.05, *p is 0.05–0.01, **p is 0.01–0.001, ***p < 0.001.**

| Net SOC storage rate in the topsoil (decadal storage rate) (ton ha$^{-1}$ yr$^{-1}$) | | Net SOC storage rate in the total soil profile (millennial storage rate) (ton ha$^{-1}$ yr$^{-1}$)[a] | |
|---|---|---|---|
| N input x soil developmental stage | | N input x cumulative soil age | |
| *Df numerator* | 1 | *Df numerator* | 1 |
| *Df denominator* | 23 | *Df denominator* | 68 |
| *F-value* | 3.39 | *F-value* | 4.15 |
| *p-value* | 0.08 | *p-value* | * |
| N input | | N input | |
| *Df numerator* | 1 | *Df numerator* | 1 |
| *Df denominator* | 24 | *Df denominator* | 68 |
| *F-value* | 47.61 | *F-value* | 17.00 |
| *p-value* | *** | *p-value* | *** |
| Soil developmental stage | | Cumulative soil age | |
| *Df numerator* | 1 | *Df numerator* | 1 |
| *Df denominator* | 24 | *Df denominator* | 68 |
| *F-value* | 33.39 | *F-value* | 93.22 |
| *p-value* | *** | *p-value* | *** |

[a] Only applicable to mature soils ($M_{NL}$ and $M_{NL}$)