# Peer review of "Icelandic grasslands as long-term C sinks under elevated N inputs"

_Biogeosciences, 2016_

## Referee Comment (RC1) · Anonymous Referee #1 · 30 Apr 2016

This is an interesting study on the effect of seabird-derived N inputs on SOC storage in N-limited grassland systems of Iceland. There are not too many studies on the long-term effect of N on ecosystem C stocks, particularly in northern environments, and this is an innovative approach to gain insight into this issue. However, there are several problematic points which have to be addressed before the paper can become acceptable. There are three main problems in this study. First of all, the study is based on the assumption that there are only N inputs introduced by the guano of seabirds (authors stated that other nutrients are not so important as the grasslands are N-limited). However, guano also contains organic and inorganic C and so the study design is biased as there is a high additional C input at seabird-sites (up to 30% of guano is organic matter!). The only way to save this study would be to measure the C content of the guano and substract it from the SOC stocks, but it is questionable if a single value for guano

C can be used for correction of total SOC stocks of the whole soil profile (probably you could estimate the guano-C-input per year, but if this can be used to correct SOC accumulation of the last 1600 years in mature soils is questionable). A further main problem is that no information on soil texture is given. Soil texture largely controls the stabilization of SOC via mineral sorption. For grassland soils Hassink (1997, Plant and Soil 191, 77-87) found are strong worldwide relationship of the maximum C storage capacity of soils with the fine mineral fraction content (medium+fine silt and clay <20 $\mu$m). Therefore, a precondition would be that the study sites are comparable in terms of soil texture. If there are differences in terms of soil texture, differences of SOC stocks could also (partly) be attributed to that. Besides, the authors stated that "subplots Mnl and Mnh were protected against possible human and livestock influence prior to the measurements. . .by enclosure cages". However, if there is livestock on Heimaey, the study sites cannot be viewed as unmanaged grasslands as on Surtsey and therefore not directly compared. Livestock grazing would not only be associated with additional manure (C and N) inputs but also with potentially enhanced decomposition of soil organic matter depending on the grazing intensity (due to animal trampling, aggregate disruption etc.). There are several minor points which have to be clarified. In terms of plant analysis, characteristics of Cerastium fontanum were investigated, as it was the only species that was present in all plots. However, it is questionable if the differences in terms of aboveground biomass etc. can solely be related to different N regimes as there were different plant communities/successional stages at the study sites. Probably, the performance of Cerastium fontanum was confounded by other more dominant species? Regarding the soil analyses and calculation of C and N stocks, it seems that the bulk density of the soils (which is necessary to calculate C and N stocks) was determined correctly, even when the authors did not use this term, but the values should be given (in g cm-3). In terms of the calculation of SOC storage rates, the millennial rates were calculated for consecutive cumulative soil ages with 200 years intervals assuming a constant soil accumulation rate in the subsoil. This assumption is too speculative: on the one hand, this is rarely the case over longer periods of time, on the other hand,

translocation of C from top- to subsoils in form of DOC may be a relevant process in this environment given high precipitation. Therefore, a calculation of SOC storage rates over the 1600 years is highly speculative and thus also the conclusions regarding the long-lasting positive effect of N inputs on C sequestration. In view of the discussion, particularly in sections 4.2.2 and 4.3, it would certainly be a benefit to include litera-ture on the C storage capacity/C saturation of soils that could also be calculated for these soils (see e.g. Hassink 1997, Wiesmeier et al. 2014, Global Change Biology 20, 653-665). After a revision in terms of these points, the manuscript can be evaluated again.

---

## Referee Comment (RC2) · Anonymous Referee #2 · 1 May 2016

This study has addressed how increased N inputs associated with seabird nesting colonies have influenced total ecosystem C stocks and net C accumulation rates in unmanaged Icelandic grassland soils. Results from this study demonstrate that increased N inputs from bird droppings (i.e. guano) are related to higher rates of soil C accumulation. I agree that greater N inputs from guano might be partly responsible for greater soil C accumulation; this is evident from absolute measurements within the high-N input sites and from the comparison between ENH and MNL sites. However, I think that the authors have chosen a rather convoluted way to show and explain their results. I think both the main emphasis of this study and the explanation of the potential underlying mechanisms responsible for higher net SOC storage rates need a major revision and key data might need to be added and analysed in order to support the main findings of this study.

1) The main emphasis of this study should simply be on the potential effects of bird droppings (guano) on net changes in soil C storage across different Iceland grasslands. The current emphasis on the "necessity for a better understanding of the N-induced stimulation of long-term C storage in northern ecosystems" is honestly outside the limits and possibilities of this study. The authors have not addressed the potential C sink ability of northern ecosystems under increasing anthropogenic N emissions but the long-term effects of guano deposition on net changes in SOC. The same authors state this in the discussion: "the N status was clearly more closely related to the annual seabird-derived N input than to ecosystem maturation", thus I would change the 'global change' emphasis from anthropogenic N inputs and ecosystem C sinks to how long term organic N inputs might influence changes in soil C accumulation rates. The introduction should set the stage of how long-term organic C and N inputs were found to be influencing C accumulation rates from previous literature studies.

I find strange that 'organic N inputs' or "N inputs from guano' are expressions never used in the manuscript.

2) If the authors aim to compare total ecosystem C and N stocks they need to clearly set a soil depth range, 0-20 or 0-30 cm for example, where total ecosystem stocks are compared across the different sites. Currently there are too many differences in ecosystem age, soil depths, and successional development stages to be able to compare main C and N stocks in a meaningful way. I think authors have to clarify what 'topsoil' means in terms of soil depth range (could this be 0-30 cm for all sites?).

3) In terms of mechanisms, a key missing factor here, which could be mainly responsible for changes in soil C accumulation rates across sites is actually the rate of C addition to soils through guano deposition. Bird droppings return both C and N to the soil with significant consequences for the formation of SOM. If the authors do not have information on rates of C additions per hectare per year, they might be missing a critical factor, which could explain as much variability as N inputs in long-term changes of soil SOC.

4) There is some confusion in relation to the species composition of the plant communities studied here. For example, on Page 6, lines 1-2: The authors state that Cerastium fontanum was the only plant species found in all experimental plots but on page 4, lines 29-30 they also state that "The MNL site hosts a species-rich grassland community, typical for low nutrient conditions (Magnússon et al., 2014)", which is contradictory and create confusion later when interpreting the results.

Other: Page 5, lines 12-15: I don't understand why sampling was done 'outside' the main long-term experimental plots (10x10 m) in "Adjacent to each permanent plot, three 0.2x0.5 m subplots were placed for destructive measurements". What is then the meaning of the permanent plots to this particular study?

Conclusion Lines 21-23, Page 12. The study does not show in any way that " the decadal net SOC storage rate of mature Icelandic grasslands was greatly stimulated by chronically elevated N inputs, which supported the theory that the increasing northern terrestrial C sink during the past decades could be (partly) caused by increasing anthropogenic N inputs". The SOC storage effect is likely due by long-term deposition of bird droppings.

Remove Fig. 6 because the positive relationship between soil C and soil N (either net changes or content %) is already well known and does not add any new insight into the main findings of this study.

---

## Author Comment (AC1) · 30 May 2016

Authors answers to the reviewers questions – Anonymous Referee #1

Comment 1:
the study is based on the assumption that there are only N inputs introduced by the guano of seabirds (authors stated that other nutrients are not so important as the grasslands are N-limited). However, guano also contains organic and inorganic C and so the study design is biased as there is a high additional C input at seabird-sites (up to 30% of guano is organic matter!). The only way to save this study would be to measure the C content of the guano and subtract it from the SOC stocks, but it is questionable if a single value for guano-C can be used for correction of total SOC stocks of the whole soil profile (probably you could estimate the guano-C-input per year, but if this can be used to correct SOC accumulation of the last 1600 years in mature soils is questionable).

**Thank you for this comment. In the submitted manuscript, we did not mention seabird C inputs as their contribution was found to be minor in comparison to the C inputs from biomass:**
**The annual C inputs from guano, calculated from the guano C/N ratio and the seabird-derived N input rates (47 and 67 kg ha$^{-1}$ yr$^{-1}$ in $E_{NH}$ and $M_{NH}$ respectively), amounted to 0.15 ton ha$^{-1}$ y$^{-1}$ on $E_{NH}$ and 0.26 ton ha$^{-1}$ y$^{-1}$ on $M_{NH}$ respectively. The annual C inputs from biomass (assuming an aboveground- and root-turnover of 1 year), amounted, however, to 12.3 and 14.4 ton ha$^{-1}$ y$^{-1}$ on $E_{NH}$ (inside the seabird colony on Surtsey) and on $M_{NH}$, respectively. The biomass C inputs have been stable over many years, especially in $M_{NH}$, which is in mature successional state, but also in $E_{NH}$, as the succession from barrens to grassland was completed in only a few years after the initiation of seabird colonization (Magnússon et al., 2014).**

**In conclusion, the guano C inputs were estimated to be 1.2 and 1.8 % of the total C inputs in $E_{NH}$ and $M_{NH}$ respectively, which is a conservative estimate, as especially the root turnover rate is expected to be higher than 1 year. Moreover, guano C inputs are more easily decomposable than biomass C inputs. Therefore, the proportion of guano C that remains in the soil after decomposition processes will be even less compared to plant-derived C.**

**We do agree that it is important to better illustrate the relative importance of guano- and biomass derived C inputs in the paper, as C sequestration is central to the study. We will therefore add this information to the 'Results' section, and will discuss it briefly in the 'Discussion' section.**

Comment 2:
No information on soil texture is given. Soil texture largely controls the stabilization of SOC via mineral sorption. For grassland soils Hassink (1997, Plant and Soil 191, 77-87) found are strong worldwide relationship of the maximum C storage capacity of soils with the fine mineral fraction content (medium+fine silt and clay <20 _m). Therefore, a precondition would be that the study sites are comparable in terms of soil texture. If there are differences in terms of soil texture, differences of SOC stocks could also (partly) be attributed to that.

**Thank you for this valuable comment.**
**Indeed, soil texture affects many soil characteristics, amongst others carbon storage capacity. We, however, experienced during our sampling effort that $M_{NL}$ and $M_{NH}$ had comparable soil textures (based on visual and manual determination). Also according to the soil classification by Arnalds (2015), both are of Brown Andosol soil type. Also $E_{NL}$ and $E_{NH}$ were experienced to have similar soil textures, at least below the main root zone. In the upper layers, $E_{NH}$ has developed an O horizon and a premature A horizon, while $E_{NL}$**

has not developed distinct soil horizons yet. In conclusion, within successional stage (early successional vs. mature), the soils were found to have similar soil texture. The difference in the upper layers between $E_{NL}$ and $E_{NH}$ is part of the treatment effect (different N input rates). We did, however, not do a full soil texture analysis using standard methods. As differences in soil texture could cause differences in C storage capacity, we will perform additional soil texture measurements during the coming 3 weeks to verify our observation that the soil texture is similar within each successional stage. We will compose a supplementary table on soil texture and dedicate a paragraph in the 'Result' section to this issue. Further, we acknowledge that soil texture differs broadly between northern grasslands (the current focus of our manuscript). Therefore, we will narrow the terminology of the manuscript to 'a subarctic grassland'.

Comment 3:
Besides, the authors stated that "subplots Mnl and Mnh were protected against possible human and livestock influence prior to the measurements by enclosure cages". However, if there is livestock on Heimaey, the study sites cannot be viewed as unmanaged grasslands as on Surtsey and therefore not directly compared. Livestock grazing would not only be associated with additional manure (C and N) inputs but also with potentially enhanced decomposition of soil organic matter depending on the grazing intensity (due to animal trampling, aggregate disruption etc.).

The additional C and N inputs from livestock into the ecosystem were minimal, as the sheep are allowed to graze the year round and did not receive additional feeding. Further, there were no indications that the grazing and manure dropping activities were separated in space, so no major redistributions of C and N were assumed.
We acknowledge that grazing can have influenced C turnover rate at the mature sites. However, the grazing pressure was similar at $M_{NL}$ and $M_{NH}$. Further, grazing was homogeneously distributed within the $M_{NL}$ and $M_{NH}$ sites, as the fertility of the sites was homogeneous (nutrient poor and nutrient rich, respectively) and therefore no preferential grazing at fertile spots took place. Consequently, we don't believe that livestock grazing did compromise comparison between these sites. Further, a large proportion of the northern grassland area is subjected to grazing. The grazing activity at the mature sites could thus increase the relevance of our findings.
Finally also the early successional sites could be called 'grazed', as graylag geese have colonized the island, feeding upon the grasslands there (Magnússon et al., 2014). In any case, the early successional site is of a very different age class as the mature sites, 26 years since grassland initiation vs ca. 1600 years, and the age/successional stage effect is expected to overrule the effect of differences in grazing pressure.

We will better indicate the similarity of the grazing pressure in $M_{NL}$ and $M_{NH}$ in the 'Material and Methods' section, to clarify their comparability in that respect.

Comment 4:
In terms of plant analysis, characteristics of Cerastium fontanum were investigated, as it was the only species that was present in all plots. However, it is questionable if the differences in terms of aboveground biomass etc. can solely be related to different N regimes as there were different plant communities/successional stages at the study sites. Probably, the performance of Cerastium fontanum was confounded by other more dominant species?

The authors have the impression that this concern is caused by a misunderstanding. We apologize for the unclarity in the manuscript.

Authors answers to the reviewers questions – Anonymous Referee #1

**What is shown in Figure 3, is the 'leaf N/P stoichiometry of *Cerastium fontanum'*, the 'plant height of *Cerastium fontanum*' and 'the total vascular aboveground biomass'. We acknowledge that the first two parameters could be confounded by interactions with other species, but we focused on this species to avoid that differences among sites were confounded by species identity. The last parameter (total biomass), however, includes all species and is not confounded by species interactions. While the separate trends (especially of N/P ratio and plant height) might not be sufficient to show N limitation, we think that the combination of these three parameters gives a strong indication that the systems are indeed N limited.**

**We will better clarify in the manuscript that Figure 3.C shows total biomass. Further, we will rephrase the paragraph about N limitation in the discussion to clarify that it is the combination of the three parameters that indicates N limitation.**

Comment 5:
Regarding the soil analyses and calculation of C and N stocks, it seems that the bulk density of the soils (which is necessary to calculate C and N stocks) was determined correctly, even when the authors did not use this term, but the values should be given (in g cm-3).

**Thank you for this comment, we will add this term to the manuscript and will express the values in g cm$^{-3}$.**

Comment 6:
In terms of the calculation of SOC storage rates, the millennial rates were calculated for consecutive cumulative soil ages with 200 years intervals assuming a constant soil accumulation rate in the subsoil. This assumption is too speculative: on the one hand, this is rarely the case over longer periods of time, on the other hand, translocation of C from top- to subsoils in form of DOC may be a relevant process in this environment given high precipitation. Therefore, a calculation of SOC storage rates over the 1600 years is highly speculative and thus also the conclusions regarding the long-lasting positive effect of N inputs on C sequestration. In view of the discussion, particularly in sections 4.2.2 and 4.3, it would certainly be a benefit to include literature on the C storage capacity/C saturation of soils that could also be calculated for these soils (see e.g. Hassink 1997, Wiesmeier et al. 2014, Global Change Biology 20, 653-665).

**Thank you for this suggestion for improvement.**
**The application of the technique proposed by Weisemeier et al. ($C_{sat - pot}$ = 4.09 + 0.37 * particles ≤ 20 μm (%)) to our dataset is, however, not straightforward. The conventional techniques for soil size fractionation are not applicable to Andosols for the following reason: Allophanes and ferrihydrates, two clay minerals that are formed during the weathering of basalt, strongly bind organic material. This results in very stable silt sized aggregates that the conventional techniques for soil size fractionation are not able to break down. This leads to an over-estimation of the silt grain size and a large underestimation of the clay grainsize, including the partiles <20 μm. However, we will perform particle size measurements and calculate the C storage potential to our best potential. We will add the suggested references to the manuscript and dedicate a paragraph in the discussion on the calculated C storage potential.**

**The soils under investigation are young soils (<1600 years) and no C saturation was expected in the upper soil layers. However, in the deeper soil layers, both $M_{NL}$ and $M_{NH}$ showed C saturation in soils older than ~1000 years (Figure 5.B). It will be interesting to see whether this saturation point agrees well with the C saturation value calculated with**

the technique of Weisemeier. We will allocate more attention to the C saturation and C storage potential the discussion.

We acknowledge that it is a simplification to assume a constant soil thickening rate between the 1973 ash layer to the 395 AD ash layer, which both gave a known age point within each soil profile. However, we think that, even if this assumption is rather imprecise, the ca. 200-year resolution in our results is still a justifiable approximation based on more detailed dating of soil profiles in S-Iceland that are not located close to active soil erosion areas (cf. Gisladottir et al., 2010). The temporal scale that could be reached in this 'natural gradient study', is not possible to acquire with controlled experiments. We therefore think that this assumption can be made, when it is transparently explained in the manuscript.

We will, however, dedicate more attention in the 'Discussion' section to the uncertainty of the assumption of a constant accumulation rate.

…………………………..

**Final comment:**

We thank you for the constructive comments and suggestions, which will greatly improve the manuscript. The authors will make a major revision of the manuscript if accepted, according to the Reviewers' comments; including e.g. a new soil texture analysis on existing samples that will be ready in ca. 2-3 weeks' time. The final revision of the manuscript text is therefore pending.

**References**

Arnalds, Ó. 2015. *The soils of Iceland, First ed.,* Dordrecht, The Netherlands, Springer.

Magnússon, B., Magnússon, S. H., Ólafsson, E. & Sigurdsson, B. D. 2014. Plant colonization, succession and ecosystem development on Surtsey with reference to neighbouring islands. *Biogeosciences,* 11**,** 5521-5537.

---

## Author Comment (AC2) · 30 May 2016

Authors answers to the reviewers questions – Anonymous Referee #2

Comment 1:
The main emphasis of this study should simply be on the potential effects of bird droppings (guano) on net changes in soil C storage across different Iceland grasslands. The current emphasis on the "necessity for a better understanding of the N-induced stimulation of long-term C storage in northern ecosystems" is honestly outside the limits and possibilities of this study. The authors have not addressed the potential C sink ability of northern ecosystems under increasing anthropogenic N emissions but the long-term effects of guano deposition on net changes in SOC. The same authors state this in the discussion: "the N status was clearly more closely related to the annual seabird-derived N input than to ecosystem maturation", thus I would change the 'global change' emphasis from anthropogenic N inputs and ecosystem C sinks to how long term organic N inputs might influence changes in soil C accumulation rates. The introduction should set the stage of how long-term organic C and N inputs were found to be influencing C accumulation rates from previous literature studies. I find strange that 'organic N inputs' or "N inputs from guano' are expressions never used in the manuscript.

**We thank the referee for this comment.**
**We agree that the present focus of the manuscript is ambitious on the issue of mineral vs. organic N inputs. However, the need for long-term N input studies on ecosystem C storage is pressing. This gap cannot be achieved with controlled mineral-N addition experiments. Natural gradients in N inputs, irrespective of N-form, such as seabird nesting areas, are one way to get a better understanding of the effects of long-term increased N inputs. Even though the design of our study is not perfect, we think that the results are still valuable in this respect.**

**However, we acknowledge that the assumption that our results are applicable for all 'northern grasslands' (which the current title suggest) is too ambitious. Therefore we narrow the focus of the manuscript to 'a subarctic grassland'. We will also include more discussion on the generalizability of our results regarding effects of anthropogenic mineral N inputs.**

Comment 2:
If the authors aim to compare total ecosystem C and N stocks they need to clearly set a soil depth range, 0-20 or 0-30 cm for example, where total ecosystem stocks are compared across the different sites. Currently there are too many differences in ecosystem age, soil depths, and successional development stages to be able to compare main C and N stocks in a meaningful way. I think authors have to clarify what 'topsoil' means in terms of soil depth range (could this be 0-30 cm for all sites?).

**At the mature study sites, the authors separated between topsoil and subsoil using the 1973 ash layer in each soil profile. This ash layer was located at 6.4 ± 0.4 (SE) and 11.4 ± 1.7 (SE) cm soil depth at $M_{NL}$ and $M_{NH}$, respectively. The soil on top of this layer (which had been accumulating over the last 40 years) contained over 70% of the roots and was the layer with the highest biological activity. At the early developmental study sites on Surtsey, the age of the whole soil profile was comparable to the topsoil of the older islands (45 years since the eruption ended in 1967). The authors therefore made no subdivision between topsoil and subsoil there.**

**We will rephrase the 'Material and Method' section about the topsoil-subsoil separation to avoid misunderstandings.**

Comment 3:

In terms of mechanisms, a key missing factor here, which could be mainly responsible for changes in soil C accumulation rates across sites is actually the rate of C addition to soils through guano deposition. Bird droppings return both C and N to the soil with significant consequences for the formation of SOM. If the authors do not have information on rates of C additions per hectare per year, they might be missing a critical factor, which could explain as much variability as N inputs in long-term changes of soil SOC.

**Thank you for this comment. We already addressed this issue in our answers to Referee 1. We do agree that it is important to show the relative importance of guano- and plant derived C inputs in the paper (guano annual C-input being ca. 1.5% of the annual plant C inputs). We will add this information to the 'Results' section, and will discuss it briefly in the 'Discussion' section.**

Comment 4:
There is some confusion in relation to the species composition of the plant communities studied here. For example, on Page 6, lines 1-2: The authors state that Cerastium fontanum was the only plant species found in all experimental plots but on page 4, lines 29-30 they also state that "The MNL site hosts a species-rich grassland community, typical for low nutrient conditions (Magnússon et al., 2014)", which is contradictory and create confusion later when interpreting the results.

**The authors have the impression that this concerns is caused by a misunderstanding. We apologize for the unclarity in the manuscript. With the expression "*Cerastium fontanum* was the only plant species found in all experimental plots", we did not mean that *Cerastium fontanum* was the only species that occurred in all experimental plots, but that this was the only species in common between all the plots. It is true that $M_{NL}$ was species rich, but $E_{NL}$ and $M_{NH}$ were species poor. The only overlapping species (over all treatments: $E_{NL}$, $E_{NH}$, $M_{NL}$ and $M_{NH}$) was *Cerastium fontanum*.**

**We will rephrase this sentence to "*Cerastium fontanum* was the only species that that was common between all the experimental plots" to avoid misunderstanding.**

Comment 5:
Page 5, lines 12-15: I don't understand why sampling was done 'outside' the main long-term experimental plots (10x10 m) in "Adjacent to each permanent plot, three 0.2x0.5 m subplots were placed for destructive measurements". What is then the meaning of the permanent plots to this particular study?

**The 'permanent survey plots' in this study were important because an extensive amount of other data has been published for these plots. They were established between 1990 and 1995 and have been followed closely ever since. To keep these 'permanent plots' undisturbed for future research, destructive soil- and vegetation sampling (as was necessary for our research) is always done outside the confines of the main plot. Placing our sampling sites close to these plots provided background information on vegetation development, seabird nesting density, soil parameters, gas exchange and many other variables.**

**To make the value of the 'permanent plots' more clear, we will dedicate a small paragraph in the 'Material and Methods' section to their long history. Further, we will change the terminology from 'permanent plots' to 'permanent survey plots'.**

Comment 6:
Page 12, Lines 21-23: The study does not show in any way that " the decadal net SOC storage rate of mature Icelandic grasslands was greatly stimulated by chronically elevated N inputs, which supported the theory that the increasing northern terrestrial C sink during the past decades could be (partly) caused by increasing anthropogenic N inputs". The SOC storage effect is likely due by long-term deposition of bird droppings.

**We are sorry that the reasoning behind this conclusion was not clear. As discussed earlier, the droppings themselves are only adding <2% of the annual C-inputs to the ecosystem. We made this conclusion because of the higher SOC accumulation rate in the topsoil (above the 1973 ash layer) of $M_{NH}$ compared to $M_{NL}$: During the past 40 years (1973 – 2013), almost twice as much SOC had been stored in $M_{NH}$ compared to $M_{NL}$. We will reword this reasoning in the 'Discussion' section, so that it is more clear why this conclusion is made:**

*Previous wording:*
**The strong positive effect of chronically elevated N inputs on decadal net SOC storage in our study is in line with the theory that the recent northern C sink is at least partly caused by increasing N deposition (Hudson et al., 1994; Lloyd, 1999; Schlesinger, 2009).**

*New wording:*
**The strong positive effect of chronically elevated N inputs on decadal net SOC storage (i.e. SOC storage in the soil layer that was accumulated after 1973), is in line with the theory that the recent northern C sink is at least partly caused by increasing N deposition (Hudson et al., 1994; Lloyd, 1999; Schlesinger, 2009).**

Comment 7:
Remove Fig. 6 because the positive relationship between soil C and soil N (either net changes or content %) is already well known and does not add any new insight into the main findings of this study.

**Thank you, this is a valid point.**

**We will move the figure to the supplementary and only discuss it briefly in the 'Discussion' section.**

……………………..

**Final comment:**

**We thank you for the constructive comments and suggestions, which will greatly improve the manuscript. The authors will make a major revision of the manuscript if accepted. This includes e.g. a new soil texture analysis requested by Reviewer 1 that will be ready in ca. 2-3 weeks' time; which is why the final revision is now pending.**

---

## Author Comment (AC3) · 13 Jun 2016

Dear editor,

Thank you very much for your positive advice and clear suggestions.

We are very pleased to be given the opportunity to revise our manuscript. We will revise the manuscript based on our previous replies to the reviewers comments. We will also take your further suggestions into account.

We hope to re-submit the revised manuscript before mid-July.

Sincerely,

Niki Leblans on behalf of all co-authors